



# Lignin's ability to nucleate ice via immersion freezing and its stability towards physicochemical treatments and atmospheric processing

Sophie Bogler[1], Nadine Borduas-Dedekind[1,2]

[1]Institute for Biogeochemistry and Pollutant Dynamics, ETH Zurich, Zurich, 8092, Switzerland
[2]Institute for Atmospheric and Climate Sciences, ETH Zurich, Zurich, 8092, Switzerland

*Correspondence to*: nadine.borduas@usys.ethz.ch, @nadineborduas

**Abstract.** Aerosol-cloud interactions dominate the uncertainties in current predictions of the atmosphere's radiative balance. Specifically, the ice phase remains difficult to predict in mixed-phase clouds, where liquid water and ice coexist. The formation of ice in these clouds originates from heterogeneous ice nucleation processes, of which immersion freezing is a dominant
pathway. Among atmospheric surfaces capable of templating ice, mineral dust, biological material, and more recently organic matter are known to initiate freezing. To further our understanding of the role of organic matter in ice nucleation, we chose to investigate the ice nucleation (IN) ability of a specific sub-component of atmospheric organic matter, the biopolymer lignin. Ice nucleation experiments were conducted in our home-built Freezing Ice Nuclei Counter (FINC) to measure freezing temperatures in the immersion freezing mode. We find that lignin acts as an ice active macromolecule at temperatures relevant
for mixed-phase cloud processes (e.g. 50% activated fraction up to – 18.8 °C at 200 mg C L$^{-1}$). Within a dilution series of lignin solutions, we observed a non-linear effect in freezing temperatures; the number of IN sites per mg carbon increased with decreasing lignin concentration. We attribute this change to a concentration-dependant aggregation of lignin in solution. We further investigated the effect of physicochemical treatments on lignin's IN activity, including experiments with sonication, heating and reaction with hydrogen peroxide. Indeed, harsh conditions such as heating to 260 °C and addition of 1:750 g of
lignin to mL of hydrogen peroxide were needed to decrease lignin's IN activity to the instrument's background level. Next, photochemistry and ozonation experiments were conducted to test the effect of atmospheric processing on lignin's IN activity. We showed that this activity was not susceptible to changes under atmospherically relevant conditions, despite chemical changes observed by UV/Vis absorbance. Our results present lignin as a recalcitrant IN active subcomponent of organic matter within for example biomass burning aerosols and brown carbon, and contribute to the understanding of how soluble organic
material in the atmosphere can nucleate ice.



## 1 Introduction

To reduce modelling uncertainties of radiative forcing from aerosol-cloud interactions, an improved understanding of ice formation in the atmosphere is necessary (Stocker et al., 2013). Atmospheric ice crystals influence cloud properties by altering their microphysical and radiative properties, thereby affecting precipitation patterns and cloud lifetime (Storelvmo et al., 2011). In fact, the majority of precipitation originates from the ice phase (Mülmenstädt et al., 2015). In mixed-phase clouds where water and ice co-exist, ice crystals grow at the expense of supercooled water droplets through the Wegener-Bergeron-Findeisen

process. As the saturation water vapour pressure over ice is lower than over water, ice crystals grow and consequently dominate the phase distribution of water in the cloud (Korolev and Field, 2008). Ice nucleating particles (INPs) are necessary to induce freezing at warmer temperatures via a heterogenous freezing pathway since homogenous freezing of cloud water droplets becomes instantaneous only at temperatures below –38 °C. In mixed-phase clouds, the immersion freezing pathway dominates heterogenous freezing and occurs when solid or dissolved INPs initiate freezing from within a supercooled cloud droplet

(Hoose and Möhler, 2012; Knopf et al., 2018). It remains crucial to study the pathways of ice crystal formation in these clouds to advance our quantitative understanding of warming and cooling factors contributing to the aerosol-cloud radiative effects (Storelvmo et al., 2011).

  In this study, we focused on the ice nucleation (IN) abilities of organic matter. Organic aerosols are ubiquitous in the

environment (Jimenez et al., 2009) and their ice nucleating ability is highly variable and depends on their chemical composition (Knopf et al., 2018). Recently, dissolved organic matter from lakes and rivers have been identified as efficient soluble INPs (Borduas-Dedekind et al., 2019; Knackstedt et al., 2018; Moffett et al., 2018). The analytical challenge, however, of resolving the chemical composition of complex organic matter hinders our ability to identify the specific functional group, chemical moieties or conformations acting as a surface to template ice. Thus, detailed elucidations of the IN active component of organic

matter, including organic aerosols, can help improve ice crystal concentration predictions in mixed-phase clouds.

  Therefore, we chose to reduce sample complexity by investigating the IN activity of a specific commercially available sub-component of organic aerosols, the biopolymer lignin. Indeed, lignin is the second most abundant organic polymer on earth after cellulose. An estimated 30% of the organic carbon present in our biosphere is part of this polymer (Boerjan et al., 2003).

Lignin functions as an essential structural component in the cell walls of plants where it builds a complex matrix with cellulose and hemicellulose (Faraji et al., 2018). As such, lignin provides stability, enables the efficient transport of water and solutes through plant stems and protects the plant against external pathogens (Boerjan et al., 2003). The biopolymer is best described as a complex class of aromatic heteropolymers built from three main precursor monomers of hydroxy-cinnamyl alcohols: namely p-coumaryl alcohol, coniferyl alcohol and sinapyl alcohol (Figure 1) (Vanholme et al., 2019). The monolignols are

most commonly linked through a stepwise and entirely chemically-controlled polymerization in irregular patterns via ether bonds (β-O-4, α-O-4) or carbon-carbon bonds (Chandra and Madakka, 2019; Ralph et al., 2019). Evidently, a variety of formation pathways exists, which likely increases the polymer's structural heterogeneity and its robustness against degradation and enables its protective role in cell wall structure.

Lignin is emitted into the atmosphere through three main pathways. (1) Lignin is part of the organic matter in lakes, rivers, and oceans as plant debris. This material enters the water body either via runoff from the watershed or through deposition, for example from overhanging plants. The production of lake, river or sea spray aerosol transfers this biological material including lignin and other biogenic macromolecules into the atmosphere (Axson et al., 2016; Knackstedt et al., 2018; Meyers-Schulte and Hedges, 1986; Olson et al., 2019; Slade et al., 2010; Zark and Dittmar, 2018). The aerosols containing this complex organic

matter have been shown to nucleate ice at temperatures relevant for mixed-phase clouds (Borduas-Dedekind et al., 2019; Knackstedt et al., 2018; Moffett et al., 2018). (2) Soils can be a source of lignin-containing organic matter to the atmosphere.


Wind erosion can transport soil dust and plant fragments into the atmosphere, for example during agricultural harvesting (Suski et al., 2018; Tobo et al., 2014). Studies from e.g. Conen et al. (2011), Hill et al. (2016), Pratt et al. (2009), Suski et al. (2018), and Tobo et al. (2014) have demonstrated the potential of this material to act as INPs before. Further, lignin could be part of

organic coatings on soil minerals known to be INPs and thereby influence the mineral's IN activity. Indeed, examinations of organic coatings on minerals have revealed the importance of the organic component (Augustin-Bauditz et al., 2016; Birkel et al., 2002; Perkins et al., 2020). (3) Lignin also reaches the atmosphere through biomass burning processes, where its monomeric pyrolysis products are common tracers of organic matter (Shakya et al., 2011; Simoneit, 2002). In addition, large fractions of polymeric forms of lignin remain present in biomass burning organic aerosols even after pyrolysis (Myers-Pigg et

al., 2016; Shakya et al., 2011; Stefenelli et al., 2019). In a study by Myers-Pigg et al. (2016), up to 73-91% of the lignin material in a wildfire smoke plume was in polymeric form. We therefore expect lignin to be present and relevant for aerosol-cloud interactions. The IN potential of these biomass burning aerosols has also been investigated before (e.g. McCluskey et al., 2014; Prenni et al., 2012), but results from field and laboratory studies remain contradictory and further elucidations require more details about particle composition and morphology (Bond et al., 2013; Kanji et al., 2017).


An assessment of the IN activity of plant-derived material including lignin and complex ambient samples has recently been initiated by Steinke et al. (2019). This study concluded that the individual plant-derived organic compounds show lower IN activity than complex ambient plant-derived samples. In fact, lignin was one of the least IN efficient components in plant-derived material with IN activity surface density values between $10^8$ and $10^9$ m$^{-2}$, i.e. 2 orders of magnitude lower than the

complex samples. With our study, we aim to continue the discussion of IN activity in plant-derived material and macromolecules in immersion freezing (Pummer et al., 2015) and contribute IN results for lignin in more detail. We aim to improve our understanding of organic matter's IN abilities on a molecular level by investigating lignin as a model organic aerosol component. To determine lignin's possible impact on aerosol-cloud interactions, we characterized its IN activity in our drop-Freezing Ice Nuclei Counter (FINC, Miller et al., 2020). Additionally, we extended the investigation to the effects of

physicochemical treatments such as sonication, heating, reaction with hydrogen peroxide, and atmospheric processing such as photochemistry and ozonation, on lignin's IN activity.

## 2 Materials and Methods

### 2.1 Lignin sample preparation

Lignin solutions were prepared from powder kraft lignin (471003, Sigma Aldrich, average $M_w$ 10 000 g mol$^{-1}$) dissolved in

molecular biology reagent water (hereafter termed *background water*, W4502, Sigma Aldrich, Germany). Targeted carbon concentrations were based on a carbon content of 50% resulting from the producer's elemental analysis data (47-51%). As specified by the producer, this technical kraft lignin has a remaining sulphur content ≤ 3.6 %. This sulphur is introduced into lignin as an impurity during the delignifying kraft process in wood pulping by addition to double bonds in its aliphatic carbon chain (Lin and Dence, 1992). Sample solutions were stored in amber glass vials at 4 °C in between processing and

measurements. Sterile conditions are important to minimize background freezing and thus all solutions were prepared in a calibrated laminar flowhood (Labculture® Class II, Type A2 Biological Safety Cabinets, ESCO). In addition, all glassware was cleaned before use by rinsing 3 times with distilled water and 3 times with acetone followed by drying in the oven at 120 °C for minimum 1 hour.



## 2.2 Analytical chemistry instruments

### 2.2.1 UV/Vis spectrometry

To characterize the chromophores within lignin, we measured the absorbance of aqueous sample solutions with UV/Vis spectrometry (Varian Cary 100 Bio, Agilent) (see Figure S14). A baseline correction with background water was applied to all sample measurements.

### 2.2.2 Ion chromatography

Acetic acid, formic acid, oxalic acid and pyruvic acid were quantified via ion chromatography (DX-320, Thermo Scientific, USA) using the method from Borduas-Dedekind et al. (2019). Briefly, we used the instrument with an EG40 eluent gradient generator, a Dionex Ion-Pac AG11-HC RFIC 4 mm column and a guard column, a Dionex AERS 500 4 mm electric suppressor and an electrical conductivity detector. The KOH gradient was set as follows: 0 to 11 min, 1 mmol $L^{-1}$; 11 to 37 min, 1 mmol $L^{-1}$ to 40 mmol $L^{-1}$; 37 to 38 min, 40 mmol $L^{-1}$; 38 to 41 min, 1 mmol $L^{-1}$. The retention times of the acids were 7.9 min, 10.6 min, 13.5 min and 26.2 min for acetic acid, formic acid, pyruvic acid and oxalic acid, respectively. The calibration curves for each acid are in the supplementary information (SI, Fig. S1). Note that none of our samples had detectable amounts of pyruvic acid.

## 2.3 Treatment Experiments

### 2.3.1 Sonication

Lignin solutions concentrated at 20 mg C $L^{-1}$ and 200 mg C $L^{-1}$ were sonicated in 10 mL volumetric flasks for up to 60 min in a sonicator (Ultrasonic Cleaner, VWR, USA) at 30°C.

### 2.3.2 Heating

Lignin was heated as a powder in 20 mg aliquots for 3 h at temperatures ranging from room temperature (21 °C) to 300 °C. After heating, background water was subsequently added to reach a carbon content of 200 mg C $L^{-1}$. It was necessary to heat lignin in powder form, as heating in solution would have led to unquantified concentration changes due to the evaporation of water. To test the ice nucleating ability of the insoluble by-products generated during the heating, vacuum filtration (0.22 $\mu$m MCE Membrane, MF-Millipore™) and sterile syringe filtration (0.22 $\mu$m PES Membrane, TPP, Switzerland) were used to separate the soluble from the insoluble heated lignin (see Figure 6).

### 2.3.3 Hydrogen peroxide reaction

The chemical treatment with hydrogen peroxide ($H_2O_2$) was adapted from Paramonov et al., 2018. Solid lignin samples reacted with $H_2O_2$ (35 wt% aqueous solution, Sigma-Aldrich) in ratios varying from 1:5 to 1:750 (g lignin : mL $H_2O_2$) in a glass vial. Using a carbon content of lignin of 50%, based on the information from the supplier, these ratios approximately equate to 1.2 times to 180 times molar excess of $H_2O_2$ to carbon. The samples were left to react overnight and then diluted accordingly with background water to reach a concentration of 200 mg C $L^{-1}$ before determining the IN activity of the solution. To ensure the reaction with hydrogen peroxide was complete and no additional change to the IN activity was introduced, one set of samples (ratios 1:5, 1:50, 1:500) was left to react for an additional time period of 3 days before dilution to 200 mg C $L^{-1}$.





### 2.3.4 Size filtration

To classify the size fraction of the lignin polymer responsible for IN activity, lignin sample solutions were filtered with sterile syringes through 0.02 $\mu$m (inorganic membrane, Whatman, Anotop) and 0.22 $\mu$m (PES Membrane, TPP, Switzerland) filters.
The filters were rinsed with 30 mL milli-Q water and 10 mL background water prior to filtration of 10 mL of sample. Freezing experiments with cellulose acetate filters (0.2 $\mu$m and 0.45 $\mu$m, VWR, USA) and a PTFE membrane filter (0.2 $\mu$m, BGB, USA) showed persistent IN activity similar to lignin's at 20 mg C L$^{-1}$ in our ice nucleation setup after rinsing (see SI, Fig. S2). Therefore, the use of these filters was discontinued for this study.

### 2.4 Atmospheric processing experiments

### 2.4.1 Photochemical experiment

For the photochemical experiments, 9 mL of a 20 mg C L$^{-1}$ lignin solution was pipetted in 10-mL borosilicate test tubes (Pyrex, 15 × 85 mm, disposable) and placed inside a motorized turn-table of a commercial photoreactor (Rayonet, Southern New England Ultraviolet Co). The photoreactor was equipped with 8 UVB light bulbs (UVB – 3000 Å from Southern New England Ultraviolet Co.) and provided an irradiation as a function of wavelength spectrum peaking at 310 nm (Fig. S3). The temperature
inside the photoreactor is kept stable with a fan at 30-32 °C for the duration of the experiment (up to 25h). At each time-point, test tubes containing sample solution were replaced with test tubes containing either more sample solution or pure milli-Q water to ensure a constant light path.

### 2.4.2 Actinometry

To quantify the light intensity in the photochemical setup, we conducted an actinometry experiment with the chemical
actinometer pyridine/*p*-nitroanisole (PNA) following the method in Laszakovits et al. (2016) and Borduas-Dedekind et al. (2019). Briefly, a solution containing 20 $\mu$M of recrystallized PNA and 0.25 mM of pyridine in nanopore water was irradiated for 6 h under the same conditions as the lignin solutions. At different timepoints, the PNA was quantified via high-pressure liquid chromatography (UltiMate 3000 HPLC, ThermoFisher Scientific). The system was equipped with a reverse-phase C18 column (Ascentis express, 90 Å C18, 15 cm x 4.6 mm, 5 $\mu$m), its guard column, and a UV detector. The analyses were
performed in isocratic mode using as eluent 50:50 A:B, where A is 100% acetonitrile (ACN) and B is 90% acetate buffer at pH 6 with 10% ACN. The eluent was delivered at a flow rate of 1 mL min$^{-1}$, while the sample injection volume was 20 $\mu$L. In these conditions, PNA eluted at 2.98 min and was detected at 316 nm. A plot of $\ln(c[PNA]/c[PNA]_0)$ versus time (SI, Fig. S4) provided the pseudo first-order degradation constant for PNA of $k_{deg,PNA} = (0.568 \pm 0.004)$ h$^{-1}$, where the error is the standard deviation of triplicate experiments. Based on $k_{deg,PNA}$ and the measured spectral irradiance of the used light bulbs, we calculated
the absolute spectral irradiance $I_\lambda$ of this photochemistry setup as $I_\lambda = (109.94 \pm 0.85)$ W m$^{-2}$ (further calculation details are described in the SI, Sect. S3). Additionally, we determined a conversion factor of 3.14 from the irradiation time in the photoreactor into the equivalent irradiation time in natural sunlight. With this factor, 25 h irradiation in the photoreactor equates to 6.5 days of sunlight in the environment, consistent with the atmospheric lifetime of organic aerosols.

### 2.4.3 Ozonation setup

Ozone (O$_3$) was generated *in situ* reaching target concentrations of 100 ppbv and 1 ppmv O$_3$. The O$_3$ concentration was monitored with an ozone monitor (932, BMT Berlin Messtechnik GmbH). In detail, compressed ambient air filtered (Druckluft-Filter, Center Diehl) from particles and volatile organic compounds (VOCs) was led with 1 L min$^{-1}$ into the O$_3$ generator (A2Z Ozone, Inc. USA) where O$_3$ was generated in a fan-cooled Corona Discharge tube from atmospheric oxygen.





The highly concentrated $O_3$-containing outflow was diluted accordingly with pure nitrogen and then bubbled for up to 6.5 h at
a flowrate of 0.1 L min$^{-1}$ through 20 mL of lignin solutions using a teflon tube. Sample solutions with lignin concentrated at
20 mg C L$^{-1}$ and background water for controls were prepared in 50 mL round-bottom flasks and stirred with a magnetic stirrer
on a stir plate throughout the ozonation experiment. The background controls ensured that no additional IN active
contamination was introduced in the ozonation setup (SI, Fig. S13). The mass loss through evaporation of solvent following
bubble bursting at the solution surface was monitored, but remained < 0.5 wt%. The effect on the final solution concentration
was therefore minor and not further considered.

### 2.5 Ice nucleation setup

We used our home-built Freezing Ice Nuclei Counter (FINC) to quantify the heterogeneous ice nucleation of aqueous lignin
through immersion freezing (Miller et al., 2020). Briefly, FINC works by using an ethanol bath to cool sample solution and
sample freezing is detected based on a change in light intensity in successive pictures captured by a mounted camera throughout
cooling. Each experiment yields 288 freezing temperatures, from 20 $\mu$L aliquots of sample solution pipetted into 288 piko
PCR tray wells. Sample trays are prepared in a laminar flow hood to minimize sources of contamination. FINC's limit of
detection is at – 23.9 ± 0.6 °C, determined based on repeated freezing experiments with background water (calculation details
in SI, Sect. S4). For lignin samples concentrated at 20 mg C L$^{-1}$ in 20 $\mu$L aliquots, FINC's uncertainty in the reproducibility is
± 0.2 °C based on one standard deviation of the $T_{50}$ values of 7 experiments (SI, Fig. S7). Additionally, we report a temperature
uncertainty of 0.5 °C in the freezing temperature of each well in FINC. This temperature uncertainty was determined based on
thermocouple calibration experiments with ethanol inside the piko PCR tray wells (Miller et al., 2020). See SI, Sect. S4 for
more details about FINC including a picture of the instrument, an exemplary recorded tray picture during a sample run and
details regarding data processing.

### 2.5.1 Normalization to organic carbon content

Freezing temperatures were normalized to total organic carbon (TOC) content following Eq. 1 according to Vali (1971, 2008,
2019). The calculation requires the frozen fraction (FF as value between 0 and 1), the concentration of non-purgeable carbon
(TOC in mg C L$^{-1}$), and the sample aliquot in each well ($V_{well}$ is 20 $\mu$L), and results in the $n_m$ value representing the ice-active
mass site density (example in Figure 2). Without the division by TOC, Eq. 1 results in the INP number concentration, also
included in Figure 2.  Note that the TOC content was calculated based on the 50 % carbon content as specified from the
vendor's elemental analysis which we assume to be constant (Sect. 2.1). Multiple attempts to quantify the TOC with a
Shidmazu instrument were unsuccessful as they led to inconsistent results likely due to inefficient combustion and
mineralization of lignin using the normal operating parameters on the TOC analyser (SI, Sect. S5).

**Eq. 1:** $n_m = - \dfrac{ln\,(1-FF)}{TOC*V_{well}}$

### 2.5.2 Data visualization

FINC data analysis was conducted in MATLAB®. Additionally to two-dimensional FF curve graphs and $n_m$, boxplots are used
to visualize freezing data. The advantage of boxplots is that one dimension is sufficient per experiment to show temperature
dependent freezing events which enables a clear and concise data presentation of experiments side by side (Figure 2) as
established in e.g. Brennan et al. (2020).





## 3. Results and literature comparison

### 3.1 Lignin as an IN macromolecule

Lignin was active as an IN macromolecule in the temperature range relevant for mixed-phase clouds. This observation led us to conduct the list of experiments visualized in Figure 3 to further characterize lignin's IN ability.

#### 3.1.1 Concentration dependence of lignin's IN ability

The 50% frozen fraction ($T_{50}$) of lignin solution ranged between $-18.8 \pm 0.15$ °C and $-22.6 \pm 0.26$ °C for a concentration between 200 mg C L$^{-1}$ (40 $\mu$mol lignin L$^{-1}$) and 2 mg C L$^{-1}$ (0.4 $\mu$mol lignin L$^{-1}$), respectively (Figure 4). Indeed, this dilution series spans two orders of magnitude where the expected decreasing trend in IN ability was observed (Figure 4a). All $T_{50}$ freezing temperatures were above the instrument's limit of detection, derived from averaging ten background water experiments (black line, Figure 4a). The frozen fraction values were then normalized by organic carbon content, known from the mass of lignin weighed while making the solutions, to obtain $n_m$ values between $-7.6$ °C to $-26.2$ °C. The $n_m$ values (Figure 4c) increased exponentially with decreasing freezing temperature and covered more than 5 orders of magnitude between 1 and $10^5$ IN sites per mg C.

Interestingly, we observe a dilution effect after normalization to organic carbon content throughout the series (Figure 4). In other words, the normalized $n_m$ values of lignin are higher for lower carbon concentrations (Fig. 4c). For example, the $n_m$ values of lignin at 2 mg C L$^{-1}$ are a factor of 10 higher than the $n_m$ values of lignin at 200 mg C L$^{-1}$. In fact, the $T_{50}$ values increased exponentially with an asymptote reaching approximately $-19$ °C with lignin concentrations higher than 50 mg C L$^{-1}$ (Fig. 4b). We interpret this result as a change in chemistry of the ice active sites of the macromolecules due to dilution effects and suspect that changes in the supramolecular structure are occurring (Sect. 4.1).

When comparing our obtained $n_m$ values for lignin's dilution series, we note that the data fits very well with the lignin parametrization developed in Miller et al. (2020) based on a 20 mg C L$^{-1}$ lignin solution. This agreement underscores the reproducibility within lignin's IN activity and FINC at the same concentration. In addition, the $n_m$ values have the same slope as the IN parametrization for biogenic particles in sea-spray aerosols from Wilson et al. (2015) (Figure 4c), but are two orders of magnitude lower than the parametrization. In comparison with $n_m$ values from river and swamp dissolved organic matter (Borduas-Dedekind et al., 2019), lignin's IN ability is also lower. Both comparisons indicate that although lignin may contribute to the IN activity in organic matter, it is not the most active component within organic matter. This finding is also consistent with Steinke et al. (2019), who identified lignin to have lower IN activity than other plant material collected from dry leaf debris from either spruce or maple trees and agricultural dust after rye and wheat harvests.

#### 3.1.2 Size dependence of lignin's IN ability

Filtering aqueous solutions of lignin through sterile syringe filters with pore sizes of 0.22 $\mu$m and 0.02 $\mu$m reduced the IN activity (Figure 5). Specifically, the $T_{50}$ freezing temperatures of the filtrate of the 200 mg C L$^{-1}$ lignin solution decreased by 0.6 °C with the 0.22 $\mu$m filter and by 1.9 °C with the 0.02 $\mu$m filter compared to the unfiltered sample. On the other hand, the 20 mg C L$^{-1}$ solutions showed a decrease of 1.3 °C in the $T_{50}$ value only after filtration through 0.02 $\mu$m. Notably, these 0.02 $\mu$m-filtered lignin solutions still showed IN activity higher than the water background by 1.3 °C, indicating that the remaining macromolecules of sizes < 0.02 $\mu$m were still active in nucleating ice via immersion freezing. In fact, all solutions after filtering remained IN active above the water background. This result illustrates how the polymers vary in size beyond the limit of the filters used and that every size bin investigated here contributes to IN activity. Thus, the decrease in IN activity after filtering





can be attributed to the loss of mass, i.e. a concentration decrease (Sect. 3.1.1), rather than to the loss of particularly active size fractions of lignin. The comparison of the IN activity of size fractions relative to each other would require the normalization to the carbon content and thus the determination of carbon concentration after filtering, which was not possible with lignin (SI, Sect. S5).

Further, we highlight the contribution of the lignin components < 0.02 $\mu$m in comparison to similar filtering experiments of INP samples collected in the field. In samples collected from the sea-surface microlayer (Irish et al., 2017; Wilson et al., 2015), IN activity was retained after filtering through 0.22 $\mu$m filters, but reduced to the background level after filtering through 0.02 $\mu$m. The ice-nucleating material was further characterized as likely biogenic. The same reduction to the background level after filtering through 0.02 $\mu$m was also observed Brennan et al. (2020), who investigated the IN ability of alpine snowmelt samples. In these studies, the INP size range was confined to 0.22 – 0.02 $\mu$m, which means lignin was not present in appreciable concentrations due to the lack of activity below 0.02 $\mu$m. However, we could now show that this lower limit of 0.02 $\mu$m cannot be generalized to describe IN macromolecules from complex organic samples universally. If IN activity remains after filtering through 0.02 $\mu$m, this activity could be attributed to fractions of lignin as a particular subcomponent of organic matter.

### 3.2 Effects of physicochemical treatments on lignin's IN ability

We investigated the effects of physicochemical treatments typically used in atmospheric ice nucleation research to deconvolute the source identities of ambient INP samples, including sonication, heating, and reaction with hydrogen peroxide. In particular, heating and the reaction with hydrogen peroxide have been used frequently to remove organic material from IN samples (e.g., (Conen et al., 2011; Hill et al., 2016; Paramonov et al., 2018; Perkins et al., 2020; Tobo et al., 2014).

### 3.2.1 Effect of sonication

Sonication is a common extraction tool to remove particulate matter from filters. As Miljevic et al. (2014) showed, this process can produce reactive radicals which may impact the aerosol's chemical composition. We therefore investigated if sonication could affect lignin's IN activity. Aqueous lignin solutions concentrated at 200 mg C L$^{-1}$ and 20 mg C L$^{-1}$ were sonicated up to 60 min. After 60 min, this treatment did not introduce a distinct change on lignin's IN activity resulting from reactive radicals produced (Fig. S8). Based on these observations, the effect of pre-treating organic aerosol samples by sonication on the subcomponent lignin's structure or its IN activity is not predicted to impact ice nucleation.

### 3.2.2 Effect of heating

Heating procedures are commonly used to remove the contributions of organic matter to IN activity in complex samples containing mixtures of heat-labile and heat-stable material. We heated dry lignin in an oven at a range of temperatures up to 300 °C for 3 h before dissolving the powder in background water. We only observed a decrease in lignin's IN activity after exposure to temperatures above 180 °C (Figure 6). In particular, at 260 °C, the IN activity was reduced to the background water level with a T$_{50}$ value of – 23.7 °C. Therefore, to completely remove the contributions from lignin to IN activity in ambient samples, a temperature above 260 °C is necessary. It is likely that when heat-stable organic fractions have been observed in complex samples after a heat-treatment < 260 °C, lignin fragments were contributing to the remaining IN activity (e.g. in Hill et al., 2016; Suski et al., 2018).

The mass of lignin was weighed before and after the heating treatment, and the weighted mass loss of lignin powder below 0.01 wt%. Thus, the observed decrease in IN activity is not due to a decrease in the total lignin concentration (Sect. 3.1.1).





With increasing heating temperatures, the lignin powder visually darkened and became insoluble. As a result, the experimental solutions contained visible suspensions of lignin from exposure to 220 °C and higher. Therefore, heating altered the polymer's chemical structure which affected the IN activity and the solubility. Indeed, these temperature-dependent structural modifications have been studied in detail previously. For example, Kim et al. (2014b, 2014a) heated milled wood lignin to between 150 °C and 300 °C and analysed modifications in the functional group composition using chromatography methods, nitrobenzene oxidation, and NMR. Briefly, the results from this study showed that from a heating temperature of 150 °C, first ether bonds connecting methoxyl groups to aromatic rings within the polymer are broken and then bonds to increasingly bigger side-chains and terminal functional groups are cleaved. Starting at 250 °C, depolymerisation occurred, including breakage of ether bonds releasing monomeric phenols. Simultaneously, condensation reactions took place, which reconnect the lignin fragments to modified polymeric structures.

To further test whether the soluble or insoluble lignin can both act as the ice nucleating macromolecule or particle, respectively, we filtered the suspensions with sterile syringe filtration using 0.22 $\mu$m PES Membrane, TPP filters and measured the IN ability of the filtrate on FINC (Figure 6). Note that attempted vacuum filtrations with 0.22 $\mu$m MCE Membrane, MF-Millipore filters led to elevated background water freezing temperatures, so their use was discontinued. We find consistently that the $T_{50}$ values of the filtrate were the same as the unfiltered heated lignin suspension (Figure 6). Therefore, the insoluble mass lost through filtering was not responsible for the remaining IN activity further confirming the role of the soluble fraction to the IN activity. Although attempts to quantify this soluble mass via TOC analysis were unsuccessful (SI, Sect. S5), we can be sure that the lignin concentration was lower in the filtrate solutions, since the filtration step removed mass. Thus, the heated and filtered soluble lignin had a higher IN activity compared to the insoluble lignin, further distinguishing IN macromolecule and IN particles. This IN activity could be due to changes in the chemical structure or due to the dilution that may change lignin's supramolecular structure in solution, as discussed in Sect. 3.1.1 and 4.1.

### 3.2.3 Effect of reactions with hydrogen peroxide

We exposed lignin to increasing concentrations of hydrogen peroxide for 24 h, then diluted the solution with background water to match 200 mg C L$^{-1}$ and subsequently measured the freezing temperatures with FINC. Additionally, we tested prolonged reaction time up to 4 days, which did not affect the freezing temperature results as illustrated in Fig. S11. This control was necessary as there were no reliable indicators to show the completion of the reaction with hydrogen peroxide even after potential bubbling, frothing, or heating of the sample has subsided (Mikutta et al., 2005; Paramonov et al., 2018). In all, lignin's IN ability decreased at a ratio of 1 g lignin : 200 mL H$_2$O$_2$ and larger (SI, Fig. S9). Thus, lignin is robust under hydrogen peroxide treatment and an exposure of at least 50 times molar excess of hydrogen peroxide to carbon is required to induce oxidative effects that lead to changes in the polymer's IN activity.

The necessary hydrogen peroxide concentration to observe change in lignin is not directly transferable to lignin's decomposition in complex IN samples. In complex ambient organic smaples, other hydrogen peroxide reaction pathways e.g. involving metal ion catalysts or other reactive intermediates are potentially available to break down the molecule (Mikutta et al., 2005). With insufficient amounts of oxidant, organic residuals may contain lignin which continues to contribute to observed IN activity after a hydrogen peroxide chemical treatment (e.g. in Paramonov et al., 2018; Suski et al., 2018). Specifically, Paramonov et al. (2018) treated their ambient soil IN samples in the ratio of 1 g material : 5 mL H$_2$O$_2$ (35 wt%). Even though the samples' carbon content was low with a maximum of 2.3%, these conditions only equate to an excess of ∼ 27 times molar excess of hydrogen peroxide to carbon. Thus, that amount of oxidant would likely not have been enough to completely oxidize lignin in the samples to remove its contribution to IN activity. We conclude that lignin's recalcitrance should be considered when developing methods to remove all organic carbon from complex IN samples.



At concentrations of hydrogen peroxide exceeding the ratio 1 g lignin : 200 mL $H_2O_2$, an odd freezing point depression from hydrogen peroxide decreased the $T_{50}$ values rapidly down to the $T_{50}$ value of a pure $H_2O_2$ background solution which was reached at the ratio of 1 g lignin : 750 mL $H_2O_2$ (SI, Fig. S9). The cause of this effect is not fully understood (SI, Sect. S7) but hinders more detailed interpretations of the freezing temperature results for this chemical treatment. Attempts to further quantify chemical changes to lignin via [1]H and [13]C NMR were unsuccessful due to lack of mass and thus signal. Further, in
UV/Vis-spectrometer measurements, lignin's signal overlapped the absorbance of $H_2O_2$ in the relevant wavelength range of 400 – 200 nm (Vaghjiani and Ravishankara, 1989; SI, Fig. S10).

**3.3 Effects of atmospheric processing on lignin's IN activity and chemical structure**

Organic aerosols have an average atmospheric lifetime of days to weeks. During this time, the aerosols will be subject to atmospheric processing, which can include photochemistry resulting from exposure to sunlight or other reactions involving
atmospheric oxidants such as ozone. Atmospheric processing causes aging in the aerosols, altering its physical and chemical properties. The changes may subsequently have an impact on their IN activity (Attard et al., 2012; Borduas-Dedekind et al., 2019; Gute and Abbatt, 2018; Kunert et al., 2019). To observe possible impacts of atmospheric processing on lignin's IN activity, we conducted photochemical experiments and reactions with ozone. To track possible structural changes, we measured UV/Vis spectra and followed the production of small low-weight organic acids resulting from photochemical decay
with IC.

**3.3.1 Effect of photochemistry on lignin's IN activity**

Lignin samples concentrated at 20 mg C $L^{-1}$ were subjected to UVB irradiation for up to 25 h in a photoreactor, corresponding to approximately 6.5 days in the atmosphere. After this photochemical exposure, the change in $T_{50}$ value was less than – 1 °C (SI, Fig. S12). Overall, we calculated a weak decreasing trend in freezing temperatures with increasing irradiation with a weak
correlation coefficient of – 0.65 (with p = 0.023). A repeated 25-h exposure experiment reproduced these results. We conclude that lignin is recalcitrant to photochemical degradation, despite its ability to act as a chromophore.

We emphasize the lack of decrease in lignin's IN activity by photochemistry as recent studies on pollen (Gute and Abbatt, 2018) and dissolved organic matter have shown otherwise (Borduas-Dedekind et al., 2019). Specifically, Gute and Abbatt
(2018) investigated the effect of indirect photochemical oxidation with the hydroxyl radical on the IN activity of birch and grey alder sub-pollen particles. This study linked the exposure to OH radicals to a decrease in IN activity of pollen. Similarly, Borduas-Dedekind et al. (2019) subjected naturally occurring dissolved organic matter to photochemical processing and found that its ability to nucleate ice is decreased at a loss rate of – 0.04 °C $T_{50}$ $h^{-1}$. In contrast, lignin's IN activity was resistant to atmospherically relevant photochemical processing and is likely to be retained throughout lignin's lifetime in the atmosphere
in the absence of other deactivating and degradation processes.

**3.3.2 Effect of ozonation on lignin's IN activity**

To simulate the atmospheric processing of lignin through ozonation, we exposed bulk solutions concentrated at 20 mg C $L^{-1}$ to a gas flow containing 100 ppbv and 1 ppmv of $O_3$ for up to 2 h. This exposure did not affect lignin's IN activity significantly (SI, Fig. S13): the $T_{50}$ values changed by 0.7 °C with 100 ppbv $O_3$ and by 0.4 °C with 1 ppmv $O_3$, which remains very close to
the uncertainty range of lignin in FINC. Furthermore, we observed no further change in $T_{50}$ during a longer exposure duration of 6.5 h with an $O_3$ concentration of 1 ppmv.



Our simplified experimental setup with bulk solutions does not recreate the conditions of oxidation by $O_3$ in the atmosphere in detail, in part due to different reaction kinetics in a cloud droplet. Still, the lack of change in freezing temperatures during
these experiments further illustrates the recalcitrance of lignin's IN activity. Tropospheric $O_3$ concentrations average around 35 – 40 ppbv globally and rarely reach 100 ppbv, even in more polluted regions (Tiwari and Agrawal, 2018). So even if faster kinetics govern the ozonation pathways in the atmosphere, an exposure equivalent to 6.5 h of 1 ppmv $O_3$ would hardly be achieved in an aerosol's lifetime. Thus, atmospheric processing by $O_3$ likely does not influence lignin's IN activity. This stability in IN activity after oxidative treatments with $O_3$ up to 1 ppm and 6 h duration was also observed by Kunert et al.
(2019) and Attard et al. (2012) who investigated fungal and bacterial ice nuclei, respectively. Notably, the oxidative gas mixtures in both these studies contained additionally nitrogen dioxide ($NO_2$) as a second oxidant in the same concentration as $O_3$.

### 3.3.3 Chemical changes in lignin through atmospheric processing

Changes in the absorption spectrum of lignin in the UV/Vis range revealed that both the photochemical processing and the
reaction with 1 ppmv $O_3$ affected the polymer's chemical structure (Figure 7). In both treatments, the absorbance decreased throughout the measured range of wavelengths with longer exposure duration, especially at the distinctive absorbance peaks around 205 nm, 230 nm, and 280 nm (SI, Fig. S14). This decrease indicates the decay of chromophores in lignin, which are mostly conjugated aromatic systems present in the monolignols (Huang et al., 2019). Interestingly, the observed decrease in absorbance matches closely in both experiments, indicating that the same functional groups were affected by photochemistry.

Furthermore, the analysis of the TOC content over time was hindered by experimental difficulties (SI, Sect. S5). We instead tracked the production of photoproducts formed from the chromophoric decay of the polymer, including formic acid, acetic acid, and oxalic acid. Indeed, these oxidation products increased in concentration with exposure duration (SI, Fig. S15). However, neither the decay of chromophores nor the resulting production of these small low-weight organic acids affected IN
characteristics (Sect. 3.3.1 and 3.3.2). Thus, the chromophoric substructures of lignin reactive to UVB light or ozone are not responsible for the observed IN activity of lignin. Alternatively, the new structures resulting from the atmospheric processing may be equally active in nucleating ice. Both conclusions underscore how lignin acts as an especially robust IN macromolecule that is recalcitrant to the effect of structural changes.

## 4 Discussion

### 4.1 Lignin's supramolecular structure in solution

The results of the dilution series (Sect 3.1.1) show that the concentration dependence of lignin's IN activity is non-linear. After normalization to carbon concentration, the $n_m$ values increase with decreasing lignin concentration (Figure 4c). We interpret this result as evidence for changes in lignin's supramolecular structure. Indeed, it has already been shown how the supramolecular shape of dissolved molecules can influence important aerosol properties including ice nucleation (Cascajo-
Castresana et al., 2020; Pfrang et al., 2017; Qiu et al., 2019). For example, light diffusion and viscosity change depending on the supramolecular structure which in turn has implications for reaction rates within the aerosol and cloud nucleation processes (Pfrang et al., 2017). Further, Cascajo-Castresana et al. (2020) showed how pH-dependent protein aggregation influences the freezing temperature regimes observed. The supramolecular structure of lignin in solution specifically is dictated by interactions with the solvent and intra- and intermolecular forces within the polymers (Huang et al., 2019; Vainio et al., 2004).
Additionally, the source of lignin, the presence of other solutes, the temperature, and the solution pH exert important influences on the supramolecular structure (Huang et al., 2019) and can lead to the presence of clusters and agglomerates (Norgren et al.,





2001; Norgren and Edlund, 2001; Vainio et al., 2004). We hypothesize that lignin is indeed aggregating in our aqueous solutions. In a cloud droplet, lignin may interact via hydrogen bonding with water helping stabilize the ice embryo (Kanji et al., 2017) and promote ice nucleation. As the concentration of lignin decreases, the probability for intramolecular interactions
within the polymer decreases. The biopolymer is less likely to form aggregates and could instead be unfolding to interact with more water molecules. Thus, at lower concentrations, the relative IN activity is increased as the additional interactions promote ice nucleation. The implication of this result is important for the interpretation of dilution series in immersion freezing experiments.

## 4.2 Use of commercial lignin

One of the caveats of our work is the use of commercially available kraft lignin as our source material. Kraft lignin is not naturally occurring lignin, but rather a by-product of the pulp and paper industry. Thus, there are differences in the structure of the lignin we used compared to native lignin in plants (Giummarella et al., 2020). Notably, every isolation method currently available introduces changes to the structure of lignin, complicating the observation of the biopolymer in its native state (Chung and Washburn, 2016; Stark et al., 2016). The kraft process in particular fragments the polymer and increases its solubility
compared to natural lignin to separate it from the pulp. Additionally, sulphur is used, which introduces thiol groups as a minor impurity into the polymeric structure. Consequently, there could be differences between the behaviour of natural and kraft lignin as IN macromolecules. Further experiments will be conducted to investigate the importance of this issue. Nonetheless, the polymer backbone structures of native and kraft lignin remain highly related and we do not expect large differences in the IN activity. We argue that native lignin would also be recalcitrant in the atmosphere and therefore has an extended lifetime
within the aerosol while retaining its ice nucleating ability in immersion mode.

## 4.3 Atmospheric Implications

Lignin is a subcomponent of organic matter in aerosols and soils and is capable of nucleating ice in mixed-phase cloud conditions. Specifically, $n_m$ values for lignin solutions ranged from 1 to $10^5$ IN sites per mg C between $-$ 7.6 °C to $-$ 26.2 °C, representing 1-2 orders of magnitude lower values than dissolved organic matter samples (Borduas-Dedekind et al., 2019;
Knackstedt et al., 2018; Moffett et al., 2018), sea surface microlayer samples (Irish et al., 2017; Wilson et al., 2015) and plant extracts (Gute and Abbatt, 2018; Steinke et al., 2019). However, lignin concentrations in the atmosphere have been estimated to be up to 150 ng m$^{-1}$ after biomass burning related events (Myers-Pigg et al., 2016). Lignin is therefore likely more abundant in the atmosphere than other plant extracts and bioaerosols, despite being less ice active.

Furthermore, lignin's IN ability shows resistance to physicochemical treatments and atmospheric processing despite structural changes observed by UV/Vis absorbance. Only harsh treatment conditions such as heating above 260 °C substantially reduced lignin's IN ability in immersion mode freezing to FINC's limit of detection. We emphasize that due to this robustness, lignin can likely be part of heat-stable components that are observed after heating treatments of complex organic INP samples. Overall, lignin's observed stability implies that lignin present in aerosols is a particularly long-lived organic component and
may contribute to the aerosols' overall IN activity throughout their atmospheric lifetime. Notably, the IN ability of lignin will change during the aerosol's lifetime, as water evaporation or condensation cycles affect the polymer's concentration and consequently its macromolecular structure.



**Data availability**

Data presented in all figures in the main text and in the supplementary information are deposited in the ETH Research Collection data repository at https://doi.org/10.3929/ethz-b-000422111.

**Author contributions**

SB and NBD designed the experiments, and SB carried them out. SB and NBD prepared the manuscript together.

**Competing interests**

The authors declare no conflict of interest.

**Acknowledgements**

We acknowledge the help of Rachele Ossola with the actinometry experiments, the technical help of Michael Rösch for FINC maintenance and the assistance of Franz Friebel in developing the ozonation setup. We also thank the NBD group members Anna Miller and Silvan Müller for insightful discussions and feedback.

**Financial support**

This research was supported by the Swiss National Science Foundation as part of an Ambizione grant (grant no. PZ00P2_179703).

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



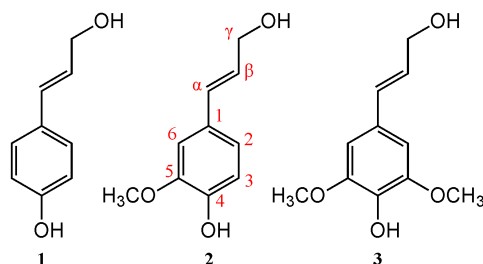

**Figure 1.** Main precursor monomers of lignin: **(1)** p-coumaryl alcohol **(2)** coniferyl alcohol **(3)** sinapyl alcohol

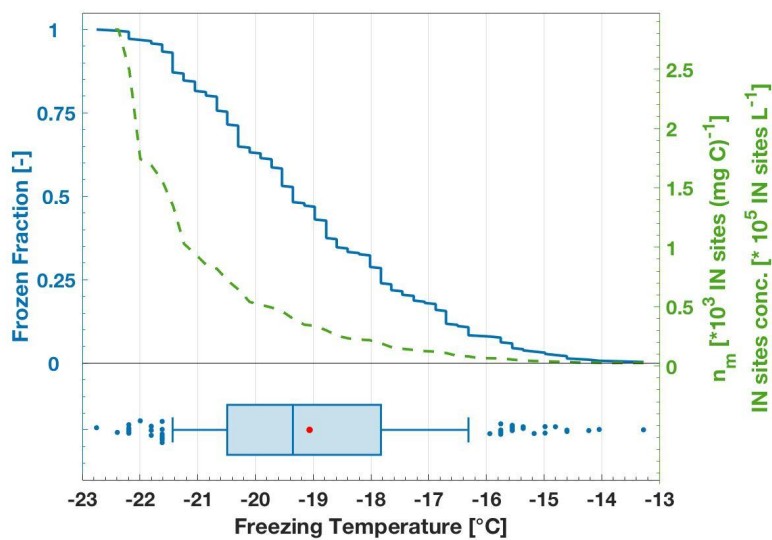

**Figure 2.** Exemplary freezing data visualization based on a 100 mg C L$^{-1}$ lignin sample solution. The left blue y-axis shows the FF curve as a stepwise function of each freezing event, 288 data points are included. The green line curve and the right green y-axis show both the INP number concentration and $n_m$ as the INP number concentration normalized to carbon content. Note that the INP conc. and $n_m$ values differ only by the factor of 100, given by the TOC concentration of 100 mg C L$^{-1}$ in this example. The FF curve on top is consolidated into a boxplot at the bottom. On the boxplot, the red dot shows the mean value, the middle blue vertical line shows the median ($T_{50}$). The box frame is limited to the 25th and 75th percentile and the box whiskers show the 10th and 90th percentile. The remaining percentiles (1st-9th, 91th-100th) are shown as individual data points with filled blue circles.



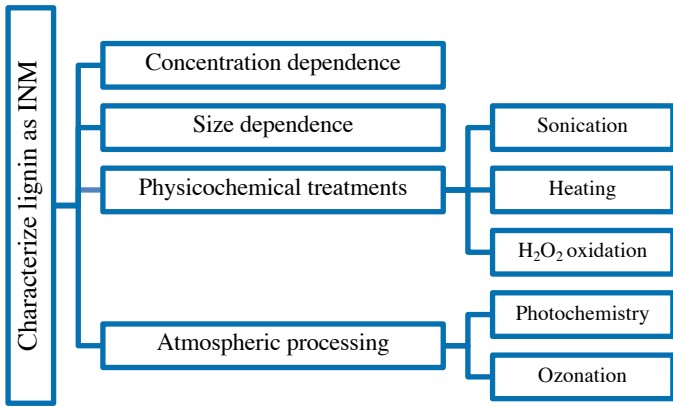

**Figure 3: Overview of conducted experiments to characterize lignin as an IN macromolecule (INM).**

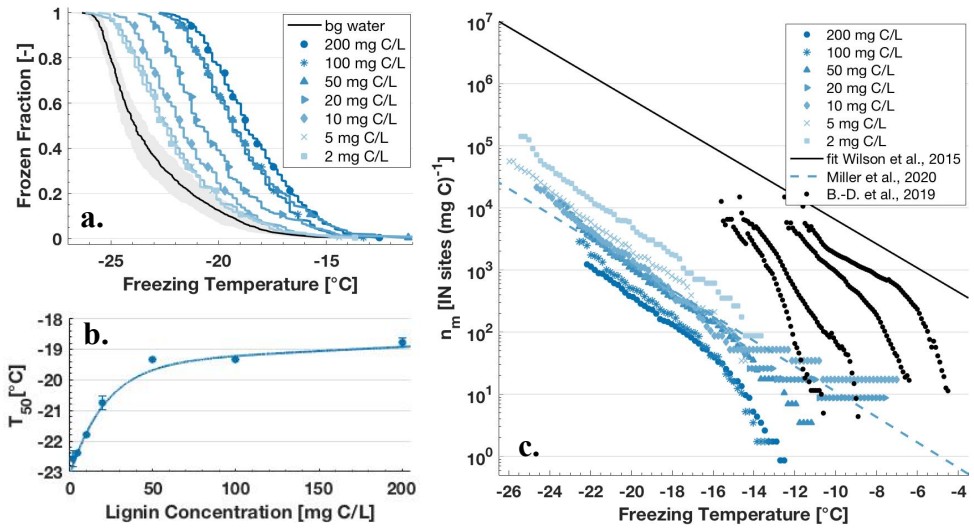

**Figure 4: (a) Freezing temperatures for aqueous dilution series as FF curves. The FF curves include the water background (bg water**
**black line) and its uncertainty range (shaded grey area) to describe FINC's limit of detection. (b) Median freezing temperature ($T_{50}$)**
**of the aqueous dilution series as a function of lignin concentration. The $T_{50}$ values include the standard deviation from multiple**
**experiments where applicable (n ≥ 2) and are fitted with $T_{50} = -3.7*exp(-0.046*x) - 19.46*exp(-0.00014*x)$, $R^2 = 0.99$. Lignin's IN**
**activity in this dilution series remains above the water background, but is decreasing exponentially with decreasing concentration**
**as illustrated by the slope of the $T_{50}$s. (c) ice-active mass site density $n_m$ as a function of temperature of the aqueous dilution series of**
**lignin. For comparison, a fit for biogenic particles in sea-spray aerosols from Wilson et al. (2015), a lignin parametrization based on**
**20 mg C L$^{-1}$ solutions from Miller et al. (2020), where $n_m = exp(-0.49*T - 1.2)$, and $n_m$ values from dissolved organic matter (Borduas-**
**Dedekind et al., 2019) are included.**





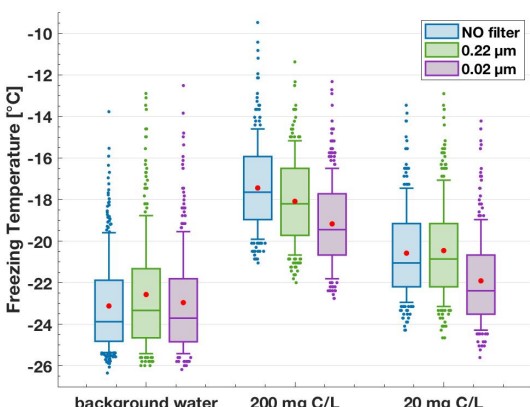

**Figure 5. Freezing temperature boxplots for the size filtration series. Aqueous lignin solutions of 200 mg C L⁻¹ and 20 mg C L⁻¹ concentration were filtered through 0.22 μm and 0.02 μm sterile syringe filters. The filtration step reduced the T₅₀ values, however the measured IN activity remained above the background for all samples.**

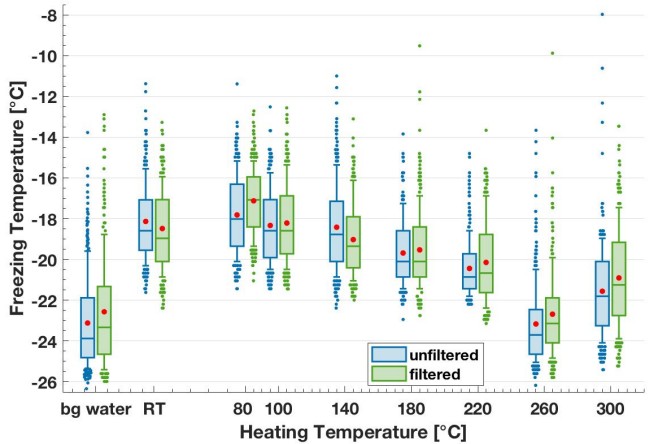

**Figure 6. Freezing temperature boxplots of the 200 mg C L⁻¹ lignin solutions directly after heating (blue, unfiltered) and after filtering through a sterile syringe filter of 0.22 μm, PES Membrane, TPP (green, filtered). *Bg water* refers to background water, *RT* refers to room temperature, i.e. 20 °C. Irrelevant of the filtering, lignin's IN activity is decreasing beginning from a heating temperature of 180 °C.**



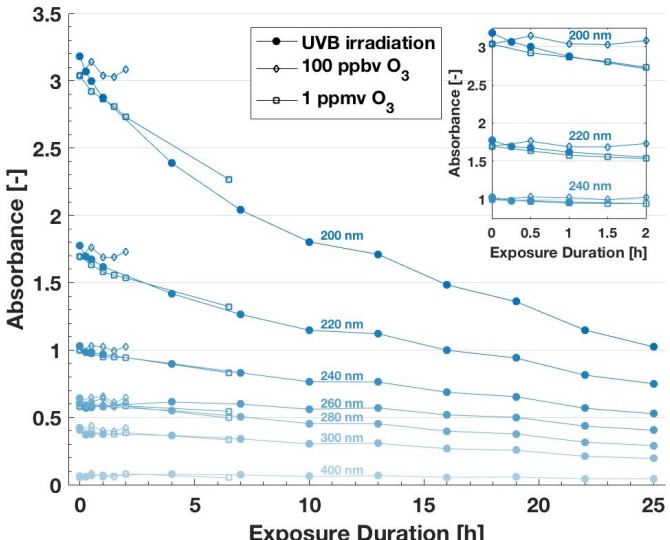

**Figure 7. Wavelength-specific absorption values from UV/Vis spectra of 20 mg C L$^{-1}$ lignin solutions after atmospheric processing through UVB irradiation and exposure to O$_3$. The insert is a zoom of the plot section from 0 to 2 h exposure duration. With increasing exposure duration, lignin's absorbance is decreasing. Changes after exposure to 100 ppbv O$_3$ are minimal.**