# Peer review of "Lignin's ability to nucleate ice via immersion freezing and its stability towards physicochemical treatments and atmospheric processing"

_Atmospheric Chemistry and Physics, 2020_

## Referee Comment (RC1) · Anonymous Referee #1 · 4 Aug 2020

The manuscript presented ice nucleation ability of a commercial lignin via immersion freezing. This study measured the frozen fraction for lignin at different carbon contents. It was shown there is a non-linear relationship between freezing temperature (T50) and lignin concentration of 2-200 mgC/L. This study also investigated the effects of sonication, heating and reaction with H2O2 and O3 on ice nucleation ability of lignin. The filtration at 0.22 and 0.2 micrometer was used to exam the size dependence of lignin's ice nucleation ability. This study provides additional data sets for the better understanding in the ice nucleation potential of lignin-like aerosol particles. Part of the

methods and conclusions need clarifications before it can be considered for publication.

Comments:

Line 100, 145, why the molecular biology reagent water was used as background water? The milli-Q water was also used during the filtration, what are the differences between these two in terms of ice nucleation measurements?

Line 180, how can ozonation approach used in this study represent the atmospheric aging by ozone? What are the potential differences or impacts?

Line 190, without detail description of FINC, it is not easy for readers to judge whether the method is appropriate. First, how the temperature uncertainty is determined? Secondly, what are the temperature differences across the whole PCR tray wells? Third, is there any correction on the data with the background water? When looking at the 2mgC/L sample, there may be over 0.3 FF contributed by the background water to 0.5 FF at T50? For 2mgC/L sample, it is about 0.2 FF. For these lower concentration samples, the background water could contribute significantly to the ice nucleation events.

Line 204, what are the estimated uncertainties in TOC concentrations and the uncertainties in ice-active mass site density (nm)? For most of the data in Fig.4C, the values of nm are well within one order of magnitude. How does it look like if uncertainties are considered? Is it still significantly different?

---

## Referee Comment (RC2) · Anonymous Referee #2 · 9 Aug 2020

This manuscript describes the ice nucleation activities of lignin after different physicochemical treatments such as sonication, heating and hydrogen peroxide digestion and simulated atmospheric processing such as photochemistry and ozone oxidation. The authors use custom-built freezing ice nuclei counter (FINC) to measure freezing temperatures in the immersion freezing mode. They also investigated effect of dilution of lignin and observe that dilution decreases frozen fraction but interestingly when the frozen fraction values were normalized by organic carbon content then active site sites per mg of carbon increases with decreasing lignin concentration. Overall, the authors found that physicochemical treatments don't not have much effect on the freezing temperatures of lignin. This manuscript can be published after appropriate revisions, mostly providing some discussion of results and elaborate some of reasoning behind the experimental design. Especially there are different treatments were performed, the authors need to justify why they did those, why they did and didn't observe any changes in freeing temperatures.

General comments: The authors selected a lignin compound that is robust against degradation and stable structure. Then several physicochemical treatments were performed. It is probably expected that there is not significant different in changes after degradation. Then why this particular lignin material was used?

Overall, discussion of the result needs to be elaborated. For example, if you didn't observe any changes in freezing temperatures after some treatment- please explain what might may cause this.

Typically, heating treatment is used to understand the effect of biological material not just to remove the contributions of organic matter. Please discuss.

Uncertainty analysis of freezing temperatures and IN active sites need to be incorporated. Please provide details about the freezing experiment set up. I was bit surprised with the detection limit of the instrument. Maybe authors should discuss limitation of this set-up. It raises concern because some of the lower concentration lignin (e.g., 2-5 mg C/L) are very close to the detection limit below -20dgree C or so. Please also discuss about normalization of carbon concentration.

Please provide some explanation, why did you use sonication? If sonication is mostly used to extract material from filter but for your experiments you have lignin in powder form. Then why didn't you see any changes in freezing temperatures?

Similarly, why the authors expected to see changes in ice nucleation activity due to decay of chromophores during simulated atmospheric processing experiments.

Can you provide little bit more information about lignin content in soils, plant debris and other sources (any quantification?) that can be aerosolize to atmosphere, this information maybe help to strengthen the atmospheric implications part.

Minor comments: Probably it is more appropriate to place Fig 3 before Fig.2

Please provide error bars in frozen fraction and active sites plot.

Figure 6: There is a decrease in freezing temperature after 260 degree C. At 260 degree C it reached already close to the background water. Then what causes a decrease in freezing temperature at 300 degree C? Probably most of the material is depolymerized at this temperature. Please explain.

---

## Author Comment (AC2) · 6 Oct 2020

We'd like to point the reviewers as well as the editor to our pre-print manuscript in ATMD describing in detail our droplet freezing technique, FINC. https://amt.copernicus.org/preprints/amt-2020-361/
* * *

---

## Author Response (AR1)

Anonymous Referee #1 Received and published: 4 August 2020

The manuscript presented ice nucleation ability of a commercial lignin via immersion freezing. This study measured the frozen fraction for lignin at different carbon contents. It was shown there is a nonlinear relationship between freezing temperature (T50) and lignin concentration of 2-200 mg C/L. This study also investigated the effects of sonication, heating and reaction with H2O2 and O3 on ice nucleation ability of lignin. The filtration at 0.22 and 0.2 micrometer was used to exam the size dependence of lignin's ice nucleation ability. This study provides additional data sets for the better understanding in the ice nucleation potential of lignin-like aerosol particles. Part of the methods and conclusions need clarifications before it can be considered for publication.

We thank the reviewer for their feedback and address the individual comments in the section below.

Comments:

Line 100, 145, why the molecular biology reagent water was used as background water? The milli-Q water was also used during the filtration, what are the differences between these two in terms of ice nucleation measurements?

 $\rightarrow$  Ice nucleation measurements are sensitive to contaminations which increase the water background's freezing temperature in FINC. Thus, we chose a very pure, commercially available water source, the molecular biology reagent water, for background measurements and as a solvent to minimize contaminations introduced by the water. The milli-Q water is less pure, results in higher background freezing temperatures in FINC and was therefore only used for a first rinsing step in line 145 to avoid wasting expensive molecular biology reagent water. The rinsing procedure was subsequently complemented by rinsing with molecular biology reagent water as well.

To add clarity, we modified the sentence which now reads, "Lignin solutions were prepared from powder kraft lignin (471003, Sigma Aldrich, average  $M_w$  10 000 g mol-1, Error! Reference source not found.) dissolved in molecular biology reagent water (hereafter termed *background water*, W4502, Sigma Aldrich, Germany) to minimize contamination from the solvent."

**Line 180, how can ozonation approach used in this study represent the atmospheric aging by ozone? What are the potential differences or impacts?**

 $\rightarrow$  We thank the reviewer for this comment. Admittedly, the oxidation setup in this study is simplified compared to atmospheric oxidation by ozone. We think the most important difference is connected to the specific reaction kinetics in the cloud droplet compared to the experimental bulk phase. Particular to our setup, the ozone bubbling in solution may be enhancing the partitioning kinetics of gaseous ozone into the solution. Nevertherless, our approach illustrates an upper exposure limit to ozone and the lack of change observed in lignin's IN ability after exposure underlines the biopolymer's recalcitrance.

In section 3.3.2 of the manuscript, we revised our argument for clarity upon receiving this reviewer's comment as follows:

"We argue that our experimental setup with bulk solutions represent an upper limit for the reactivity of lignin towards  $O_3$  in the atmosphere. The bubbling of a flow of  $O_3$  within the solution is not directly comparable to atmospheric gas phase  $O_3$  partitioning. Indeed, bubbling supplies a larger water-air interface thereby increasing the partitioning of gaseous  $O_3$  into solution, leading to higher dissolved  $O_3$  concentrations in our experimental setup. Nevertheless, the lack of change in freezing temperatures during these experiments further illustrates the recalcitrance of lignin's IN activity. Tropospheric  $O_3$  concentrations average around 35 - 40 ppbv globally and rarely reach 100 ppbv, even in more polluted regions (Tiwari and Agrawal, 2018). Thus, atmospheric processing by  $O_3$  likely does not influence lignin's IN activity. This stability in IN activity after oxidation by  $O_3$  up to 1 ppm and 6 h duration was also observed by Kunert et al. (2019) and by Attard et al. (2012) who investigated fungal and bacterial ice nuclei, respectively. Notably, the oxidative gas mixtures in both these studies contained additionally nitrogen dioxide (NO2) as a second oxidant in the same concentration as  $O_3$ , and still no significant effect was observed. [...]"

**Line 190, without detail description of FINC, it is not easy for readers to judge whether the method is appropriate. First, how the temperature uncertainty is determined? Secondly, what are the temperature differences across the whole PCR tray wells?**

→ We thank the reviewer for addressing this caveat and agree that the description of our ice nucleation setup with FINC is brief and concise. However we are happy to refer the reader to the pre-print manuscript under review by Miller et al., 2020 in Atmospheric Measurement Techniques Discussions titled "Development of the drop Freezing Ice Nuclei Counter (FINC), intercomparison of droplet freezing techniques, and use of soluble lignin as an atmospheric ice nucleation standard" (Manuscript No.: amt-2020-361) which addresses the open questions about FINC in extensive detail.

Note to reviewer and editor: A link to the manuscript will be available shortly (it has passed pre-review and technical validation) and will be linked to this paper's discussion.

Third, is there any correction on the data with the background water? When looking at the 2mgC/L sample, there may be over 0.3 FF contributed by the background water to 0.5 FF at T50? For 2mgC/L sample, it is about 0.2 FF. For these lower concentration samples, the background water could contribute significantly to the ice nucleation events.

→ There is no correction with background water. We followed the recommendation by (Polen et al., 2018) and by our own FINC manuscript (pre-print Miller et al. 2020 in AMTD). Briefly, the frozen fraction represents a cumulative probability of freezing. The nature of a probability however prevents the use of a subtraction correction which has incorrectly been applied in the past to remove the contribution of background freezing. Thus, we show the whole frozen fraction curve for both background water and sample which does indeed include an overlap for the 2 mg C/L and 5 mg C/L with the one standard deviation uncertainty range of the background water. Here, we admit that measurements of these lower lignin concentrations are at the detection limit of our ice nucleation setup and the freezing contributions from the water background and sample solution cannot be fully disentangled. Therefore, we chose the higher concentration of 20 mg C/L for the subsequent experiment series. We also refer the reviewer to the next point for a detailed discussion of sources of error.

Line 204, what are the estimated uncertainties in TOC concentrations and the uncertainties in ice-active mass site density (nm)? For most of the data in Fig.4C, the values of nm are well within one order of magnitude. How does it look like if uncertainties are considered? Is it still significantly different? We thank the reviewer for this comment. We discuss in the supplementary information, Section S5, that the TOC analysis in our solutions was challenging with our available instrumentation. We therefore relied on the supplier's specification of carbon content (~ 50%). Since we submitted this manuscript, we have also been able to confirm this value independently with an acid digestion of lignin followed by TOC analyzer measurement. We can report TOC concentration uncertainties as a sum of (1) the uncertainty of the balance (+/- 0.01 mg) and (2) volumetric flask (+/- 0.06 mL). The variability in nm introduced by these sources is insignificant and does not affect our conclusions of the observed spread in the dilution series.

Additionally, to test the limits of the spread in  $n_m$  we observed, we added Figure S8 which considers the effect of a hypothetical maximum TOC variability of 50 % on lignin's  $n_m$  values in the dilution series. Even then, the spread in  $n_m$  remains significant compared to the error.

To further clarify these uncertainties, we have now added error bars to Fig.4c, we have added Figure S8 and we reworded the discussion in Section 2.5.2 as follows:

"[...] Without dividing by TOC, Eq. 1 results in the number of IN sites, also plotted in **Error! Reference source not found.** Note that the TOC content was calculated based on the 50 % carbon content by mass as specified from the vendor's elemental analysis (Sect. 2.1). Uncertainties in the TOC content were quantified based on sample preparation, and included the balance ( $\pm$  0.01 mg) and the volumetric flask ( $\pm$  0.06 mL) and illustrated as error bars in Figure 4c. An additional discussion of errors related to  $n_m$  can be found in the SI (Sect. S5). [...]"

**Anonymous Referee #2 Received and published: 9 August 2020**

This manuscript describes the ice nucleation activities of lignin after different physicochemical treatments such as sonication, heating and hydrogen peroxide digestion and simulated atmospheric processing such as photochemistry and ozone oxidation. The authors use custom-built freezing ice nuclei counter (FINC) to measure freezing temperatures in the immersion freezing mode. They also investigated effect of dilution of lignin and observe that dilution decreases frozen fraction but interestingly when the frozen fraction values were normalized by organic carbon content then active site sites per mg of carbon increases with decreasing lignin concentration. Overall, the authors found that physicochemical treatments don't not have much effect on the freezing temperatures of lignin. This manuscript can be published after appropriate revisions, mostly providing some discussion of results and elaborate some of reasoning behind the experimental design. Especially there are different treatments were performed, the authors need to justify why they did those, why they did and didn't observe any changes in freezing temperatures.

We thank the reviewer for their feedback and address the individual comments both directly below and by revising sections in the manuscript.

General comments: The authors selected a lignin compound that is robust against degradation and stable structure. Then several physicochemical treatments were performed. It is probably expected that there is not significant different in changes after degradation. Then why this particular lignin material was used?

→ We thank the reviewer for this attentive comment. Indeed, at the beginning of our study we expected to observe more degradation upon processing lignin. This expectation was based on the results from e.g. Borduas-Dedekind et al., 2019 and Gute and Abbatt, 2018 who processed dissolved organic matter (DOM) and showed how the complex material degraded after atmospheric processing. We knew lignin is part of DOM and wanted to follow-up these studies by processing this subcomponent in depth and observe its specific behavior. As the reviewer is pointing out, we now present results that underline how lignin is stable and robust against degradation by physicochemical and atmospheric processing.

Still, we see our findings as relevant because establishing the biopolymer's stability strengthens lignin's atmospheric relevance. If removal from the atmosphere other than by wet and dry deposition is hard to achieve, lignin will have a long atmospheric lifetime and contribute to the overall ice nucleating activity throughout that time.

In the text, we added a sentence as the end of the introduction to highlight how our initial hypothesis had to be revised. The text now reads,

"Although we expected lignin, as a subcomponent of organic matter, to be reactive towards these processing treatments, we found that it was particularly recalcitrant."

Overall, discussion of the result needs to be elaborated. For example, if you didn't observe any changes in freezing temperatures after some treatment- please explain what might may cause this.

 $\rightarrow$  We thank the reviewer for this comment and hope to address it fully by specifying where we see the source of stability within the polymer. Indeed, the stability likely results from the lack of labile functional groups in lignin's structure which limits the polymer's reactivity (we added a figure of the polymer to figure 1). For example, the carbon backbone of the polymer is neither is carbon-based and is strong and the esters and ether groups would require harsh acid hydrolysis to be broken apart. On the other hand, the chromophoric, aromatic substructures are subject to a reaction in the photochemical processing we applied, as shown by a decrease in UV/Vis absorbance. However their decay did not result in changes in freezing temperatures. Evidently, these specific substructures are either not the source of lignin's ice nucleating ability or the products after photochemical processing are equally active in nucleating ice.

Further, upon receiving this reviewer's comment we have now strengthened our argument in section 4.3. as follows: "[...] Only harsh treatment conditions such as heating above 260 °C substantially reduced lignin's IN ability in immersion mode freezing to FINC's limit of detection. We link the source of lignin's stability to the lack of labile functional groups in its structure. For example, the carbon backbone of the polymer is strong, and the ester and ether groups would require harsh acid hydrolysis to be broken apart. We emphasize that due to this robustness, lignin can likely be part of heat-stable components that are observed after heating treatments of complex organic INP samples. [...]"

Additionally, as an illustration, we have revised Figure 1 in the manuscript which now includes a polymeric exemplary structure of lignin based on the structure provided by our supplier Sigma Aldrich. This figure is mentioned in lines 63, 65, 112 and 614.

**Typically, heating treatment is used to understand the effect of biological material not just to remove the contributions of organic matter. Please discuss.**

 $\rightarrow$  We think this question can be clarified by our definition of organic matter as a generic term that includes biological material. So any treatment that removes organic matter in general would also target biological material.

To further add clarity, we have revised the introduction, section 3.2 and section 3.2.2 in the text: "In this study, we focused on the ice nucleation (IN) abilities of organic matter. We define organic matter as a generic term for material that is made of mostly C, H, O, N and S covalently bonded which includes biological material as a specific subset. [...]", "[...] In particular, heating and the reaction with hydrogen peroxide have been used frequently to remove organic material including biological material from IN samples [...]", "[...] Heating procedures are commonly used to remove the contributions of organic matter including biological material to IN activity in complex samples containing mixtures of heat-labile and heat-stable material. [...]"

**Uncertainty analysis of freezing temperatures and IN active sites need to be incorporated. Please provide details about the freezing experiment set up. I was bit surprised with the detection limit of the instrument. Maybe authors should discuss limitation of this set-up.**

→ We thank the reviewer for addressing this caveat and agree that the description of our ice nucleation setup with FINC is very brief and concise. However we are happy to refer to the newly submitted manuscript by Miller et al., 2020 in Atmospheric Measurement Techniques titled "Development of the drop Freezing Ice Nuclei Counter (FINC), intercomparison of droplet freezing techniques, and use of soluble lignin as an atmospheric ice nucleation standard" (Manuscript No.: amt-2020-361) which addresses the open questions about FINC in extensive detail. The manuscript has now gone through the pre-review and should be posted momentarily. We will post a link to the manuscript describing the FINC validations and detection limit as soon as possible.

It raises concern because some of the lower concentration lignin (e.g., 2-5 mg C/L) are very close to the detection limit below -20dgree C or so. Please also discuss about normalization of carbon concentration.

→ There is no correction with background water. Instead, we decided to show the whole frozen fraction curve for both background water and sample which does indeed include an overlap for the 2 mg C/L and 5 mg C/L with the one standard deviation uncertainty range of the background water. Here, we admit that measurements of these lower lignin concentrations are at the detection limit of our ice nucleation setup and the freezing contributions from the water background and sample solution cannot be disentangled fully at the lower frozen fractions. Therefore, we chose the higher concentration of 20 mg C/L for the subsequent experiment series.

→ We thank the reviewer for addressing the discussion on normalization of carbon content. Upon receiving this comment we have revised our analysis as follows. First, we have modified the nm calculation to include all 288 recorded freezing data points which optimizes the representation of ice nucleating activity. Then, we have added a discussion of uncertainty in TOC content. As discussed in the supplementary information, Section S5, the TOC analysis in our solutions was very challenging with our available instrumentation. We therefore relied on the supplier's specification of carbon content (~ 50%). This limits our ability to report uncertainty in the TOC concentrations to sources of uncertainty during the solution preparation, i.e. the uncertainty of the balance (+/- 0.01 mg) and volumetric flask (+/- 0.06 mL) used. The variability in nm introduced by these sources proved not to be large enough to significantly alter the discussion of the observed spread throughout the dilution series.

Nevertheless we have now added the detailed uncertainty values to Fig.4c and introduced the calculation in Section 2.5.2 as follows:

"[...] Note that the TOC content was calculated based on the 50 % carbon content as specified from the vendor's elemental analysis which we assume to be constant (Sect. 2.1). Uncertainties in the TOC content were quantified based on sample preparation, and included the balance ( $\pm$  0.01 mg) and the volumetric flask ( $\pm$  0.06 mL) and illustrated as error bars in Figure 4c. An additional discussion of errors related to  $n_m$  can be found in the SI (Sect. S5). [...]"

Lastly, to test the limits of the spread in  $n_m$  we observed, we have added Figure S8 that considers the effect of a hypothetical maximum TOC variability of 50 % on lignin's  $n_m$  values in the dilution series. Even then, the spread in  $n_m$  remains significant compared to the error.

**Please provide some explanation, why did you use sonication? If sonication is mostly used to extract material from filter but for your experiments you have lignin in powder form. Then why didn't you see any changes in freezing temperatures?**

 $\rightarrow$  We thank the reviewer for these questions and hope to answer them fully as follows: in our physicochemical sections we included some of the most common tools for pre-treating atmospheric samples, among which is sonication. Sonication is a widely used tool for lab work and in the atmospheric community specifically for extracting filter material. We agree that our sample preparation did not include the extraction of lignin from filters. However, our experiment in solution established in general that lignin is not reactive towards the radical pathway with hydroxy radicals that can be formed upon sonication (Miljevic et al, 2014) and the lack of reactivity results in a lack of change in freezing temperatures.

Refer to section 3.2.1. where we clarified our statement as:

"Based on these observations, the effect of pre-treating or extracting organic aerosol samples using sonication is predicted to have no impact on lignin's IN activity."

**Similarly, why the authors expected to see changes in ice nucleation activity due to decay of chromophores during simulated atmospheric processing experiments.**

 $\rightarrow$  We thank the reviewer for this comment. A decrease in ice nucleation after photochemical processing has previously been linked to the structural decay of chromophores in dissolved organic matter (DOM) e.g. by Borduas-Dedekind et al., 2019. For our study, we have extended the investigation to include

oxidation by ozone as alternative atmospheric processing pathway that may affect ice nucleation activity. As lignin is a subcomponent of DOM that contains chromophores, we expected it to behave similarly to the complex DOM. Additionally, the chromophores are a central structural component, so their loss may have an impact on the overall structure. These structural properties may be the source of ice nucleation activity.

To further clarify this question, we have added to the following lines in the manuscript, section 3.3.: "[...] Atmospheric processing causes aging in the aerosols, altering its physical and chemical properties. For example, photochemical processing causes degrades chromophores. These changes may subsequently have an impact on their IN activity (Attard et al., 2012; Borduas-Dedekind et al., 2019; Gute and Abbatt, 2018; Kunert et al., 2019). [...]"

Can you provide little bit more information about lignin content in soils, plant debris and other sources (any quantification?) that can be aerosolize to atmosphere, this information maybe help to strengthen the atmospheric implications part.

→ We thank the reviewer for this question which has identified an open point of research. There is a lack of detailed quantifications of lignin fractions in atmospheric aerosols on a global scale. Still, there is sufficient evidence for lignin's presence and relevance in the atmosphere. In particular, biomass burning events are important point sources where concentrations up to 150 ng lignin m-3 have been measured (Myers-Pigg et al., 2016). The current wildfire events in e.g. North America (09/2020) and the long-range transport of this smoke throughout the continent underline the growing importance of this source.

Furthermore, estimates for atmospheric plant debris exist based on measurements of atmospheric cellulose as a tracer. For example, Sánchez-Ochoa et al. (2007) reported annual average concentrations of plant debris between 33.4 and 363 ng m-3 depending on the sampling location. Puxbaum and Tenze-Kunit (2003) reported an average of 750 ng m-3 over a time series of 9 months at an urban sampling location. As lignin and cellulose are two highly related biopolymers in terms of their sources, we think these quantifications are a first, valid upper estimate for concentration ranges of lignin.

We have added this information to the section 4.3. as follows: "[...] However, lignin concentrations in the atmosphere have been estimated to be up to 150 ng m-3 after biomass burning related events (Myers-Pigg et al., 2016). Lignin is therefore likely more abundant in the atmosphere at certain time-points than other plant extracts and bioaerosols, despite being less ice active. Another quantitative estimate for lignin's relevance in the atmosphere is based on quantification of atmospheric plant debris. For example, Sánchez-Ochoa et al., 2007 reported annual average concentrations of plant debris between 33.4 and 363 ng m-3 depending on the sampling location. (Puxbaum and Tenze-Kunit, 2003) reported an average of 750 ng m-3 of plant debris and of 374 ng m-3 of cellulose over a time series of 9 months at an urban sampling location. As lignin and cellulose are related biopolymers, these values may provide an upper limit for a concentration range of atmospheric lignin. [...]"

**Minor comments: Probably it is more appropriate to place Fig 3 before Fig.2**

 $\rightarrow$  We thank the reviewer for this suggestion and have adapted the order of Figure 2 and 3 accordingly.

Please provide error bars in frozen fraction and active sites plot.

→ We thank the reviewer for this comment. The error bars in the frozen fraction figures report the uncertainty in freezing temperature of each well, that results from the uncertainty in the bath and the spread over 3 trays, which is constantly +/- 0.5 °C for both sources specific for FINC (Miller et al., 2020). To avoid cluttering in the figures we have opted to report this uncertainty only in the written text (Section 2.5) and have now further added the value to the caption in Figure 4a. To illustrate this decision, here is the frozen fraction plot with added uncertainty at each 1/288 step of freezing:

Regarding the active sites plot, we refer the reviewer to our answer to the comment: "Please also discuss about normalization of carbon concentration."

Figure 6: There is a decrease in freezing temperature after 260 degree C. At 260 degree C it reached already close to the background water. Then what causes a decrease in freezing temperature at 300 degree C? Probably most of the material is depolymerized at this temperature. Please explain.

→ We thank the reviewer for addressing this caveat in our discussion. As the freezing temperatures continuously decrease with increasing heating temperatures higher 180 °C we expected this trend to include the highest heating temperature of 300 °C as likely more and more material is depolymerized. Instead however we observed a turning point of this decreasing trend at 300 °C and admittedly, we have no explanation for this observation at this point. Still, we decided to be transparent and show the dataset of the whole measurement range to create room for further discussion and interpretation ideas that can be followed up by further experiments.

To address this specific caveat more openly in the manuscript, we have added the following sentences to section 3.2.2.: "[...] Therefore, to completely remove the contributions from lignin to IN activity in ambient samples, a temperature above 260 °C is necessary. It is likely that when heat-stable organic fractions have been observed in complex samples after a heat-treatment < 260 °C, lignin fragments were contributing to the remaining IN activity (e.g. in Hill et al., 2016; Suski et al., 2018). Of note, we observed an unexpected increase of the  $T_{50}$  value to -21.8 °C for a heating temperature of 300 °C, challenging the decreasing trend in IN activity. However, the reason for this increase at 300 °C remains unclear. [...]"

**Note to editor:**

Since the original manuscript was submitted, the method development in quantifying the carbon content in the lignin solution with the available instrumentation has improved (SI, Sect. S5). With information on the carbon content newly available, we saw a potential impact on the interpretation of our results specifically for the photochemical processing section. This is why we re-ran those experiments and adapted our discussion in Section 3.3.3 accordingly: "[...] Interestingly, the observed decrease in absorbance matches closely in both experiments, indicating that the same functional groups were affected by photochemistry. The preliminary analysis of the TOC content (SI, Sect. S5) showed a potential average decrease of 25% by weight after 25 h of UVB irradiation. However, we know that this decrease is not large enough to substantially influence our interpretation of ice nucleating activity after normalization to carbon content (SI, Fig. S8). Furthermore, we tracked the production of photoproducts formed from the chromophoric decay of the polymer, including formic acid, acetic acid, and oxalic acid. [...]"

[revised manuscript text omitted]

| Formatted: Font: Tim
grammar | nes, 10 pt, Not Bold, Check spelling and |
|---------------------------------|------------------------------------------|
| Deleted: Fig. 1                 |                                          |
| Formatted: Font: Tim            | nes, 10 pt                               |
| Deleted: exemplary po           | lymeric structure in                     |
| Formatted: Font: Tim            | nes, 10 pt                               |
| Formatted: Font: Tim
grammar | nes, 10 pt, Not Bold, Check spelling and |

[revised manuscript text omitted]

**2.5.3 Data visualization**

260

FINC data analysis was conducted in MATLAB®. Additionally to two-dimensional FF curve graphs and  $n_m$ , boxplots are used to visualize freezing data. The advantage of boxplots is that one dimension is sufficient per experiment to show temperature dependent freezing events which enables a clear and concise data presentation of experiments side by side (Figure 3) as established in e.g. Brennan et al. (2020).

**3. Results and literature comparison**

**3.1 Lignin as an IN macromolecule**

Lignin was active as an IN macromolecule in the temperature range relevant for mixed-phase clouds. This observation led us to conduct the list of experiments visualized in Figure 2, to further characterize lignin's IN ability.

**265 3.1.1 Concentration dependence of lignin's IN ability**

The 50% frozen fraction (T50) of lignin solution ranged between –  $18.8 \pm 0.15$  °C and –  $22.6 \pm 0.26$  °C for a concentration between 200 mg C L-1 (40  $\mu$ mol lignin L-1) and 2 mg C L-1 (0.4  $\mu$ mol lignin L-1), respectively (Figure 4). Indeed, this dilution series spans two orders of magnitude where the expected decreasing trend in IN ability was observed (Figure 4a). All T50 freezing temperatures were above the instrument's limit of detection, derived from averaging ten background water

- 270 experiments (black line, Figure 4a). The frozen fraction values were then normalized by organic carbon content, known from the mass of lignin weighed while making the solutions, to obtain  $n_m$  values between 7.6 °C to 26.2 °C. The  $n_m$  values (Figure 4c) increased exponentially with decreasing freezing temperature and covered more than 5 orders of magnitude between 1 and 105 IN sites per mg C.
- P75 Interestingly, we observe a dilution effect after normalization to organic carbon content throughout the series (Figure 4). In other words, the normalized nm values of lignin are higher for lower carbon concentrations (Fig. 4c). For example, the nm values of lignin at 2 mg C L-1 are a factor of 10 higher than the nm values of lignin at 200 mg C L-1. In fact, the T50 values increased exponentially with an asymptote reaching approximately 19 °C with lignin concentrations higher than 50 mg C L-1 (Fig. 4b). We interpret this result as a change in chemistry of the ice active sites of the macromolecules due to dilution effects and suspect that changes in the supramolecular structure are occurring (Sect. 4.1).

When comparing our obtained  $n_m$  values for lignin's dilution series, we note that the data fits very well with the lignin parametrization developed in Miller et al. (2020) based on a 20 mg C L-1 lignin solution. This agreement underscores the reproducibility within lignin's IN activity and FINC at the same concentration. In addition, the  $n_m$  values have the same slope

285 as the IN parametrization for biogenic particles in sea-spray aerosols from Wilson et al. (2015) (Figure 4c), but are two orders of magnitude lower than the parametrization. In comparison with  $n_m$  values from river and swamp dissolved organic matter (Borduas-Dedekind et al., 2019), lignin's IN ability is also lower. Both comparisons indicate that although lignin may contribute to the IN activity in organic matter, it is not the most active component within organic matter. This finding is also consistent with Steinke et al. (2019), who identified lignin to have lower IN activity than other plant material collected from

290 dry leaf debris from either spruce or maple trees and agricultural dust after rye and wheat harvests.

| 1  |    |   |    |   |   |
|----|----|---|----|---|---|
| ι. | na | 0 | 00 | • | 2 |
|    | ve |   |    |   | ~ |

| (Formatted: Font: Times, 10 pt |
|--------------------------------|
| Deleted: Fig. 4                |
| Formatted: Font: Times, 10 pt  |
| Deleted: Fig. 4                |
| Formatted: Font: Times, 10 pt  |
| Deleted: Fig. 4                |
| Formatted: Font: Times, 10 pt  |
| Deleted: Fig. 4                |
| Formatted: Font: Times, 10 pt  |
| Deleted: Fig. 4                |

**300 3.1.2 Size dependence of lignin's IN ability**

[revised manuscript text omitted]

**9**

[revised manuscript text omitted]

| Deleted: ozonation                                                                              |                                 |
|-------------------------------------------------------------------------------------------------|---------------------------------|
| Deleted: through                                                                                |                                 |
| Deleted: ozonation                                                                              |                                 |
| Deleted: :                                                                                      |                                 |
| Deleted: which remains very                                                                     |                                 |
| Deleted: close                                                                                  |                                 |
| Deleted: range of lignin                                                                        |                                 |
| Deleted: longer                                                                                 |                                 |
| Deleted: O                                                                                      |                                 |
| Deleted: simplified                                                                             |                                 |
| Deleted: does not                                                                               |                                 |
| Deleted: recreate the conditions of                                                             |                                 |
| Deleted: oxidation by                                                                           |                                 |
| Deleted: in detail                                                                              |                                 |
| Deleted: , in part due to different reaction droplet. For example, the partitioning kine | on kinetics in a cloud
etics |
| Deleted: are affected as the surface inter                                                      | rface in our bulk setup with    |
| Deleted: A                                                                                      |                                 |
| Deleted: bigger                                                                                 |                                 |
| Deleted: surface                                                                                |                                 |
| Deleted: directly                                                                               |                                 |
| Deleted: es                                                                                     |                                 |
| Deleted: potential transitioning                                                                |                                 |
| Deleted: rate                                                                                   |                                 |
| Deleted: ozone                                                                                  |                                 |
| Deleted: .                                                                                      |                                 |
| Deleted: Still                                                                                  |                                 |
| Deleted: So even if faster kinetics gover                                                       | rn the ozonation pathways2]     |
| Deleted: oxidative treatments with                                                              |                                 |
| Formatted                                                                                       | [3]                             |
| Field Code Changed                                                                              | [4]                             |
| Formatted                                                                                       | [5]                             |
| Formatted                                                                                       | [6]                             |
| Formatted                                                                                       | [7]                             |
| Deleted: Fig. 7                                                                                 |                                 |
| Deleted: w-%                                                                                    |                                 |
| Deleted: could show                                                                             |                                 |
| Deleted: amount of a                                                                            |                                 |
| Deleted: .                                                                                      |                                 |
| Deleted: the analysis of the TOC conter                                                         | t over time was hindered        |

[revised manuscript text omitted]

---

## Author Response (AR2)

Authors note to the editor concerning the "author's certification" statement

**Re: Small change to Figure 4c since the acceptance of our ACP manuscript**

The results presented in this manuscript were obtained using our home-built drop Freezing Ice Nuclei Counter (FINC) instrument which is part of a method development and intercomparison study currently under review in AMTD (amt-2020-414) entitled, "Development of the drop Freezing Ice Nuclei Counter (FINC), intercomparison of droplet freezing techniques, and use of soluble lignin as an atmospheric ice nucleation standard" co-authored by Anna J. Miller, Killian P. Brennan, Claudia Mignani, Jörg Wieder, Robert O. David, and Nadine Borduas-Dedekind.

Since the acceptance of our ACP manuscript, we identified an issue with one of the intercomparison instruments, leading to a small change in the amt-2020-414 manuscript (Miller et al. (2020)) parameterization slope which we also depict in Figure 4c of our ACP manuscript. The slope changed from $n_m=\exp(-0.49*T-1.2)$ to $n_m=\exp(-0.558*T-3.12)$. The equation of the parameterization has now been updated in our manuscript and in Figure 4c. This parameterization does not change our conclusions in any way, but we wanted to be transparent and communicate to the editor that our current manuscript submission has this small change in Figure 4c.

Sincerely,

*Nadine Borduas-Dedekind*

Nadine Borduas-Dedekind, PhD

SNSF Ambizione principal investigator
Institute for Biogeochemistry and Pollutant Dynamics
Institute for Atmospheric and Climate Science
ETH Zürich
Universitätstrasse 16
8092 Zürich, Switzerland
Office: CHN O 16.2
Phone: +41 44 632 7315
E-Mail: nadine.borduas@usys.ethz.ch
Twitter: @nadineborduas
Group webpage: www.atmoschemgroup.org